# High Doses of Botulinum Toxin Type A for the Treatment of Post-Stroke Spasticity: Rationale for a Real Benefit for the Patients

**DOI:** 10.3390/toxins14050332

**Published:** 2022-05-06

**Authors:** Andrea Santamato

**Affiliations:** Neurorehabilitation, Spasticity and Movement Disorders Unit, Policlinico Riuniti, University of Foggia, 71122 Foggia, Italy

**Keywords:** post stroke spasticity, botulinum toxin type A, higher doses

## Abstract

In the past few years, there was a great interest in the use of higher doses of botulinum toxin type A, especially in case of upper and lower limb severe spasticity. To date, only one prospective, non-randomized, single-arm, multicenter, open-label, dose-titration study with the employment of incobotulinum toxin up to 800 U has been published, and the authors investigated safety and tolerability. Other researches showed efficacy in spasticity reduction, but there is a lack of evidence about the reasons to use high doses of botulinum toxin. This short communication highlights the benefits of higher doses for subjects with upper and lower limb spasticity.

## 1. Introduction

An increasing and cumulative body of evidence suggests that botulinum toxin type A (BoNT-A) represents the gold standard therapy for focal spasticity and other several neurological disorders. Since 1989, the effectiveness of BoNT-A in reducing post-stroke spasticity showed safety, effectiveness, reversibility and low prevalence of complications [1,2,3].

BoNT has clearly been recommended as first-line treatment for focal spasticity by several European consensus statements and the American Academy of Neurology [4,5] and current guidelines suggested the employment of a dose up to 600 units (U) of onabotulinumtoxinA (Botox^®^, Allergan, Inc., Irvine, CA, USA) and incobotulinumtoxinA (Xeomin^®^, Merz Pharmaceuticals GmbH, Frankfurt, Germany) or up to 2500 U of abobotulinumtoxinA (Dysport^®^, Ipsen, Slough, UK/Galderma, Paris, France) per injection session to treat spasticity after stroke [5].

In the past few years, there was a great interest in the use of higher doses, especially in case of upper and lower limb severe spasticity [6,7]. However, it is important to be careful regarding the risk of adverse effects related to the spread of toxin distant from the site of injection: botulism-like syndrome, exaggerated muscle weakness, dysphagia, breathing or speech difficulty [8].

## 2. High Doses of BoNT-A for the Treatment of Post-Stroke Spasticity

However, what do higher doses mean? Higher doses do not mean the increase of the dose pro muscle, but the increase of the number of the injected muscles using the conventional doses.

In addition, what are the reasons to use higher doses of BoNT-A to treat spasticity after stroke? The aim of the management of patients with upper motor neuron syndrome is to reduce the impact of spasticity and to prevent secondary complications. A correct evaluation of the patient to be injected is necessary to identify the objective of the treatment.

It is important to understand that the role of BoNT-A in reducing spasticity due to several disorders has been modified, changing from simple muscle chemodenervation to become a useful tool for the needs of the patients, such as improving limb posture, applying splint, consenting hygiene and reducing pain, permitting the treatment of standing and walking patients with spastic equino-varus foot deformities.

It is known that controversy exists regarding the improvement in motor function relative to the improvement in spasticity; low doses of BoNT-A can be used to try to increase motor function in those patients affect by spasticity graded 1 or 2 as measured by the Modified Ashworth Scale (MAS). Instead, in the case of severe spasticity, with MAS ≥ 3, higher doses can be used to reach other clinical objectives.

Paresis of the arm and spasticity of the muscle tone are the main causes of a motricity reduction; spasticity has a neurogenic component (increase in velocity-dependent, tonic stretch reflexes) and rheological changes such as stiffening and shortening of the muscle and other soft tissue (muscle and soft tissue contraction, tendon retraction, fat and fibrosis). Therefore, even if BoNT-A acts on a neurogenic component, paresis and rheological changes (present especially in severe spasticity) could impede the movement recovery of the arm, especially for upper limbs with MAS ≥ 2. Consequently, biomechanical changes in the muscle and soft tissue often contribute to spasticity more than stretch hyperreflexia [9,10].

It is for this reason that higher doses injected into the muscles of upper limb do not permit us to obtain a full recovery of the movement; in fact, higher doses injected reduce the arm spasticity, but, as cited above, the active movement of finger flexion and extension is not recovered, both regarding the weakness due to BoNT-A in the same muscles injected and for muscle paresis; different results are obtained in cases of severe spasticity of the lower limb, requiring higher doses to reduce plantar flexors’ muscles, The equino-varus pattern is due to medial and lateral gastrocnemius, soleus, tibial, finger flexor and flexor hallucis muscle hyperactivity; these lower limb muscles are mainly treated with BoNT-A (increasing totally the BoNT-A doses). The spasticity reduction obtained after the employment of botulinum toxin therapy in more muscles, and the improvement of the all-foot contact, and furthermore without a selective control of dorsiflexion ankle movement, reduces the fall risk, increasing balance and spatio-temporal gait parameters [11].

In cases of severe spasticity, the increase of BoNT-A doses obtained when clinicians inject more spastic muscles, instead of injecting only few muscles, could determine a great reduction in spasticity; this therapeutic effect is similar to a flaccid muscle condition that makes passive movement easier, permitting us to dress the subject treated, especially for spasticity affecting the upper limb.

So, even if the active movement of the hand is not possible, the possibility to inject high doses into the finger and wrist flexor muscles permits us to open the clenched fist and to reduce thumb-into-palm clenching, applying a splint or orthosis to facilitate hygiene, and cut the nails and provide skin care; otherwise, these are very difficult with finger and wrist flexor muscle MAS ≥ 3.

Moreover, the increase in doses in more spastic muscles of a joint (instead of an increase into only one muscle) could determine a great reduction of spasticity of a limited joint that could appear with a great passive movement as a flaccid muscle condition that makes it easier to dress the subject treated, especially spasticity affecting the upper limb.

The use of an ultrasound guide during the BoNT-A injection permits us to view the treatment’s muscle area and, in the case of fat and fibrosis replacement, it is possible to increase the dose (and dilution) for a greater spread of the toxin in the identified muscle, with a greater link at neuro-muscle junctions.

In addition to the importance of considering the treatment of more spastic muscles of a joint, increasing the cumulative dose to reduce spasticity impeding the movement is an option to increase the articular range of motion with passive mobilization during rehabilitative programs with a positive effect on muscular stretching, thus reducing the tendon’s retraction of the treated upper or lower limb.

As reported above, although controversy exists regarding the improvement in motor function relative to spasticity reduction after BoNT-A treatment, it is important to consider not only the motor impairment and loss of dexterity, but also clinical phenomena associated, such as pain, disability and discomfort sensation, that can be reduced with high doses of BoNT-A injection.

Spasticity-related pain is at rest or during passive or active movements and is one of the common causes of the rehabilitation failure [12].

Several papers describe the positive effects on pain and unpleasant sensation (heaviness, rigidity) in subjects affected by spasticity treated with BoNT-A injection [13].

Subjects with spasticity experience two types of pain: muscle pain due to muscle fixed contracture and compressed blood vessels with prolonged ischemia that cause a lack of oxygen which releases excitatory chemical mediators that sensitize nociceptors, and neuropathic pain due to cerebral lesions.

In both cases, BoNT-A acts not only in reducing muscle contracture and rigidity sensation, but also in reducing the release of glutamate, substance P and calcitonin-gene-related peptide that work as excitatory chemical mediators increasing the sensitization processes on the afferent fibers pathway [14].

Many times, the subjects treated with BoNT-A refer to have experienced a positive effect, including when there was not a great improvement in the motricity or spasticity reduction. Most likely, this subjective sensation is related to the reduction of pain, rigidity and heaviness, so higher doses could be used especially for cases of great pain, considering both causes of pain in stroke survivors.

Finally, considering this issue related to the employment of high neurotoxin doses, the reason for its use must be carefully indicated. High doses can be employed to reduce severe spasticity improving hygiene, pain, posture, gait and balance. A correct guide of injection, such as electrical stimulation, electromyography or ultrasonography, is needed to reduce the possibility of diffusion to near tissue and to identify the correct muscles increasing the precision of needle insertion in small and deep-seated muscles and to reduce the risk of failure of the therapy. To the author’s knowledge, no studies showed the BoNT-A concentration in blood vessels after the injection, to predict the diffusion and, consequently, the spread of BoNT-A in other areas, instead of the treated muscles; for this reason, instrumental guides do not permit to exclude the drug from completely entering small blood vessels, so it is necessary to publish research works measuring the concentration of botulinum toxin in the blood after the injection procedures. This is very important to reduce the risks of side effects and to increasing the efficacy of the treatment.

Moreover, the evidence coming from the few studies available reviewing the safety of elevated BoNT-A doses in treating post-stroke spasticity should be treated cautiously.

In fact, it is important to choose an individual primary functional target before starting, adjusting the dose to the aim of the spasticity treatment ranging from low to high doses if necessary.

As well as, it is possible to try to reduce post-stroke spasticity with the employment of BoNT-A and extracorporeal shock wave therapy (ESWT) to find a balance between cost and medical expense. Several published research works show a great effect both on hypertonia and rheological changes into spastic muscles, both increasing the antispastic effect if combined with botulinum toxin type A therapy [15,16,17].

## Data Availability

Not applicable.

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
