# Peer review of "High Doses of Botulinum Toxin Type A for the Treatment of Post-Stroke Spasticity: Rationale for a Real Benefit for the Patients"

_toxins, 2022, doi:10.3390/toxins14050332_

Round 1
Reviewer 1 Report
The author proposed point of views regarding the higher dose BoNT-A use in managing spasticity. Here are some suggestions list below:
Abstract
Line 9 “botulinum toxin up to 800 U”, please clarify to the specific BoNT-A , e.g., nabotulinumtoxin A.
Keyword
I suggest change the term “spasticity after stroke” to “post-stroke spasticity”
Introduction
Line 29 “However be careful respect to the risk of adverse effects related to spread”, please revise for the unclear sentence.
Line 30 Are you meaning of “botulism-like syndrome”
Line 46-47, 48-49
please consider revising the punctuate of the sentence
Line 51-52” In fact the reduced motricity in subjects affected by stroke is related to paresis in the arm and spasticity;” Unclear meaning of the sentence.
Line 53 Beyond the biomechanical component, some article consider “rheological change” contribute to the spasticity, or more precisely speaking “muscle hypertonia”. I suggest encompass rheological change into the paragraph.
Line 59-64
please consider revising the punctuate of the sentence for better understanding
Line 60-62
I acknowledge that the author trying to explain the higher dose of BoNT-A injection have no role in motor recovery. However, I suggest the sentence can be condensed.
Line 63-69, 74-77
please also consider revising the punctuate of the sentence for better understanding
Line 78-80
The author mentioned that when muscle has fibrotic or fatty change along with time, higher BoNT-A dose is needed. However, because of the BoNT-A target to the NMJ of the preserved muscles. In my point of view, moving the needle in the targeted muscle instead of push all the BoNT-A into same site under ultrasound guidance, is enough to bring clinical benefit. I would question the necessity to increase the dose in such “rheological” change of muscle hypertonia.
The author provided comprehensive consideration of the higher dose of BoNT-A use, however as we know, there are growing treatments other than BoNT-A to manage the spasticity. For example, extracorporeal shockwave therapy was proposed for decades, and showed the efficacy by meta-analysis (DOI: 10.1177/0269215520932196 ; DOI: 10.1080/10749357.2019.1654242) and network meta-analysis (DOI:https://doi.org/10.1016/j.eclinm.2021.101222). Higher dose of botulinum toxin could have increased medical expenditure for the stroke survivors. In my point of view, combination with emerging treatment and find a balance between cost and medical expense is important.
Author Response
please see the attachment pdf file

Reviewer 2 Report
Line 34. A typo, the article "the" twice.
Lines 110-114. I agree that all of these precautions should be taken when using existing botulinum toxin preparations. But I would point out that an inherent property of all existing botulinum toxin preparations on the market is their ability to diffuse and have a systemic effect. Therefore, modern pharma faces the challenge of developing such immobilized "adhesive" botulinum toxin preparations that would give medical professionals a tool that can be used accurately, safely, and with predictable efficacy, including the applications described by the authors of this article.
I would like to add that even such "glue" promising botulinum toxin preparations have difficulties in their application. And if, for example, in preclinical trials we can inject under visual control to completely exclude the drug from entering small blood vessels, then in subsequent clinical application we would need extraordinary efforts to achieve this using noninvasive control methods.
I would also like to add that it is worth mentioning the necessary studies to measure the concentration of botulinum toxin in the blood after such procedures. I believe that mere observation of the induced effects is not enough to fully understand the risks and effectiveness of this procedure. It is true that the manufacturers of botulinum toxin preparations have not performed pharmacokinetic studies, but this was acceptable with the state of the art when they brought the preparations to market and with the doses they suggested using. Today, the state of the art is much more advanced and much higher doses of botulinum toxin are proposed, which requires an expanded research protocol.
Author Response
please see the attachment pdf file

Round 2
Reviewer 1 Report
The author had revised the manuscript well. I have no further comment.